# Positive Tetrahydrocurcumin-Associated Brain-Related Metabolomic Implications

**DOI:** 10.3390/molecules28093734

**Published:** 2023-04-26

**Authors:** Slavica Josifovska, Sasho Panov, Nikola Hadzi-Petrushev, Vadim Mitrokhin, Andre Kamkin, Radoslav Stojchevski, Dimiter Avtanski, Mitko Mladenov

**Affiliations:** 1Faculty of Natural Sciences and Mathematics, Institute of Biology, Ss. Cyril and Methodius University, 1000 Skopje, North Macedonia; 2Department of Physiology, Pirogov Russian National Research Medical University, Ostrovityanova Street, 1, 117997 Moscow, Russia; 3Friedman Diabetes Institute, Lenox Hill Hospital, Northwell Health, 110 E 59th Street, New York, NY 10022, USA

**Keywords:** tetrahydrocurcumin, curcumin, brain injury, Alzheimer’s disease, Parkinson’s disease, mitochondria, reactive oxygen species, antioxidants

## Abstract

Tetrahydrocurcumin (THC) is a metabolite of curcumin (CUR). It shares many of CUR’s beneficial biological activities in addition to being more water-soluble, chemically stable, and bioavailable compared to CUR. However, its mechanisms of action have not been fully elucidated. This paper addresses the preventive role of THC on various brain dysfunctions as well as its effects on brain redox processes, traumatic brain injury, ischemia-reperfusion injury, Alzheimer’s disease, and Parkinson’s disease in various animal or cell culture models. In addition to its strong antioxidant properties, the effects of THC on the reduction of amyloid β aggregates are also well documented. The therapeutic potential of THC to treat patterns of mitochondrial brain dysmorphic dysfunction is also addressed and thoroughly reviewed, as is evidence from experimental studies about the mechanism of mitochondrial failure during cerebral ischemia/reperfusion injury. THC treatment also results in a dose-dependent decrease in ERK-mediated phosphorylation of GRASP65, which prevents further compartmentalization of the Golgi apparatus. The PI3K/AKT signaling pathway is possibly the most involved mechanism in the anti-apoptotic effect of THC. Overall, studies in various animal models of different brain disorders suggest that THC can be used as a dietary supplement to protect against traumatic brain injury and even improve brain function in Alzheimer’s and Parkinson’s diseases. We suggest further preclinical studies be conducted to demonstrate the brain-protective, anti-amyloid, and anti-Parkinson effects of THC. Application of the methods used in the currently reviewed studies would be useful and should help define doses and methods of THC administration in different disease conditions.

## 1. Background

Chemoprevention, generally defined as the use of natural food chemicals and/or synthetic substances to slow, inhibit, block, or even reverse the progression of human diseases, is a relatively new technique for preventing degenerative diseases in humans. Tetrahydrocurcumin (THC), as a significant metabolite of curcumin (CUR) (derived from the roots of *Curcuma longa Linn*.), has been shown to possess antioxidant, anti-inflammatory, neuroprotective, and anti-cancer properties. In this review, we analyze the existing data and the underlying molecular mechanisms of the neuroprotective properties of THC, as well as its potential implications for the prevention of different brain-related diseases.

## 2. The Structural Feature of THC Associated with Its Antioxidant Properties

THC includes phenol and β−diketone functional groups, which are common structural characteristics of antioxidant compounds. (Figure 1). In this direction, by exposing it to peroxyl radicals, Sugiyama et al. [1] found that THC produced four oxidation products derived from the β−diketone. Moreover, Wu et al. [2] described the breaking of the C–C bond in the β−diketone that occurs during redox reactions, which means that the structure of the β−diketone plays a key role in the antioxidant properties of THC [1].

In vivo, studies show that THC has a stronger antioxidant effect than CUR. THC lowered the levels of lipid peroxidation markers in the blood, liver, and kidney of cholesterol-fed rabbits [3]. THC’s antioxidant activity was also beneficial in reducing chloroquine-mediated damage in the rat kidneys by augmenting the endogenous non-enzymatic and enzymatic antioxidants and inhibiting lipid peroxidation [4,5,6]. In the same direction, Nakmareong et al. [7] showed that administration of a THC-containing diet in a rat model of N(omega)-Nitro-L-Arginine Methyl Ester (L-NAME)-induced oxidative stress leads to a significantly reduced production of superoxide (O_2_·) and malondialdehyde (MDA), followed by increased endogenous synthesis of glutathione (GSH) [8]. Similarly, THC significantly reduced L-NAME-induced aortic wall thickness and stiffness [9]. Ma et al. [10], investigating the relationship between the antioxidative brain potential of brain tissue and cognitive impairment in a C57BL/6 mouse model induced by acute hypobaric hypoxia, discovered that THC improved cognitive impairment, accompanied by reduced oxidative stress and increased glucose transporter 1 (GLUT1) protein levels. In addition, one crucial brain-related THC-affected mechanism is the synthesis of the deacetylase, sirtuin 1 (Sirt1) [11]. Sirt1’s activity is associated with improved cellular physiological function and is considered to have an anti-aging effect. Sirt1 promotes the production of brain-derived neurotrophic factors, which is one of the most significant brain-related effects. [12]. For these reasons, practical measures that might boost Sirt1 activity are of considerable interest. Among the few already-proven nutraceuticals that have potential in this regard, THC is one of the most prominent. THC was found to increase Sirt1’s mRNA as well as the levels of the protein, but the details about how THC accomplishes this remain obscure [13,14].

Several human diseases, including aging, diabetes, neurodegeneration, and cancer, have been linked to oxidative stress as one of their most prominent causes [15,16]. THC may have the ability to prevent oxidation-related human diseases due to its significant antioxidant activity, which has already been demonstrated in many in vitro and in vivo settings [17]. On the other hand, from a pharmacokinetic point of view, THC, compared to hexahydrocurcumin, for instance, has lower pharmacokinetic properties and lower bioavailability in various relevant models [18]. Based on its kinetic solubility, metabolic stability, gastrointestinal (GI) and blood–brain barrier (BBB) penetration properties, and lipophilic-ligand efficiency, THC is not at the top in comparison to some other curcuminoids [18]. Nevertheless, taking into account its advantages, such as in the case of the promotion and activation of Sirt1, considerable emphasis in this systemic review will be given to the impact of THC on neurodegenerative onset. However, its protective role in all previously mentioned diseases cannot be excluded due to the systemic relationships between them.

## 3. THC-Related Neuroprotective Effects in Hippocampal HT22 Cells

Several previous studies have shown that, as a result of its antioxidant properties, THC can prevent neuronal cell death during traumatic brain injury [19,20]. Thus, it has been shown that THC can reduce glutamate-induced death of hippocampal HT22 cells [21]. To test the neuroprotective effect of THC on glutamate-induced oxidative stress, Park et al. [21] exposed HT22 cells to 5 mmol/L glutamate in the presence or absence of THC for 24 h. The obtained data showed that glutamate decreased cell viability, while THC significantly increased cell viability at doses of 10 and 20 mmol/L compared to cells treated only with glutamate. Considering that oxidative stress has a significant role in neuronal cell death, suppression of reactive oxygen species (ROS) can be considered a potential method for slowing down neuronal cell death. Based on this, the fact that THC significantly reduces the accumulation of intracellular ROS induced by glutamate treatment represents a key step in neuroprotection [22]. Considering that an increase in intracellular Ca^2+^ ([Ca^2+^]_i_) is characteristic of neuronal cell death caused by glutamate-induced oxidative stress [23,24], Park et al. [21] measured [Ca^2+^]_i_ levels in HT22 cells and found that THC causes significant suppression of glutamate-induced [Ca^2+^]_i_ accumulation. These findings imply that THC may protect HT22 cells from glutamate toxicity by inhibiting oxidative stress and preventing [Ca^2+^]_i_.

Previous studies have implied that glutamate induces apoptotic cell death, followed by cell necrosis [25,26]. While examining the effect of THC on glutamate-induced apoptotic damage in HT22 cells, Park et al. [21] found in their study that chromatin condensation, a morphological marker of apoptotic cell death [27], is significantly increased in HT22 cells treated with glutamate, but THC completely prevents such effects. In addition, the same authors investigated whether the inhibition of mitogen-activated protein kinase (MAPK) phosphorylation (as a mechanism responsible for cell survival) underlies the prevention of glutamate-induced apoptosis [21]. The obtained results show that inhibiting intracellular ROS causes inhibition of MAPK phosphorylation and cell death induced by peroxide (H_2_O_2_) generation, indicating that ROS-mediated MAPK phosphorylation is involved in neuronal cell apoptosis [28,29]. It was found that glutamate increases the stimulation of c-Jun N-terminal kinase (JNK), extracellular signal-regulated kinase (ERK), and p38, whereas THC significantly decreases glutamate-induced phosphorylation of MAPK [21]. These findings imply that inhibition of MAPK phosphorylation is the molecular sword of THC-mediated neuroprotection (Figure 2).

## 4. THC-Related Neuropathic Protection

Current research has shown that mice injected with vincristine develop chemotherapy-induced peripheral neuropathy (CIPN) [30,31]. In the study by Greeshma et al. [32], rats injected with vincristine were characterized by lower motor nerve conduction velocity, functional loss (lower sciatic functional index), elevated oxidative stress, and TNF-α production in the sciatic nerve. It was shown that THC treatment significantly improved the nociceptive threshold in vincristine-injected rats while reducing oxidative stress, inflammatory mediators, and total [Ca^2+^]_i_ levels in the sciatic nerve. THC treatment also showed a protective effect (dose-dependent) on the decline of the functional index and conduction velocity induced by vincristine [32].

Vincristine generally causes hyperresponsiveness of A-δ and C-fiber nociceptive neurons, which sensitize dorsal horn neurons, causing hyperalgesia and allodynia [33]. According to published data, spinal microglia and astrocytes react to vincristine-induced peripheral neuropathy [34]. Thus, it was reported that activated glial cells secrete upregulated pronociceptive mediators such as nitric oxide (NO), prostaglandins, pro-inflammatory interleukins, and TNF-α [34]. Hence, any drug that suppresses pronociceptive and pro-inflammatory mediators is a potential suppressor of neuropathic pain [35,36]. THC’s analgesic and anti-inflammatory effects underlie the suppression of vincristine-induced peripheral neuropathy [34]. Additionally, THC is superior to CUR in reducing the activation of inducible NO synthase (iNOS), nuclear factor kappa light chain enhancer of activated B cells (NF-κB), cyclooxygenase 2 (COX-2), JNK, and ERK (Figure 3) [37].

Another model of clinical pain, formalin-induced nociception, occurs as a result of tissue damage and is characterized by an acute (0–10 min) and a delayed (20–40 min) phase [38,39]. THC treatment induces an analgesic response only in the delayed phase, indicating that it can block inflammatory mediators in the process of causing pain [38].

It is also known that the accumulation of [Ca^2+^]_i_ ions induces secondary messengers (calpain and calmodulin), which may further be the causes of axonal degeneration. In vincristine-treated rats, intrathecal injection of Ca^2+^ chelators dramatically reduces allodynia and hyperalgesia [40]. On the other hand, the suppressive capacity of THC upon the [Ca^2+^]_i_ ions in the sciatic nerve could be taken as a reason that defines its protective role against vincristine-induced peripheral neuropathy (Figure 4).

Finally, all the above-presented findings show that THC attenuates vincristine-induced biochemical, neurophysiological, and histological changes in rats. The benefits of THC may be associated with various mechanisms, including antinociceptive, anti-inflammatory, Ca^2+^-accumulation-inhibitive, TNF-α suppression, neuroprotective, and antioxidant activities [40].

## 5. THC-Related Induction of Mitochondrial Apoptotic Route, Autophagy, and PI3K/AKT Pathways: Neuroprotection after TBI and I/R Injury

The capacity of THC to enhance the activity of endogenous antioxidant enzymes potentiates its antioxidant properties [41,42,43]. It has been established that the increase in ROS after traumatic brain injury (TBI) causes oxidative stress, a disorder caused by subsequent brain damage [44]. During respiration, mitochondria are known to generate ROS, which can be further used to oxidize proteins and DNA [45]. At the same time, if there is damage to the mitochondria, ROS accumulation interferes with mitochondrial function and disrupts the balance of the redox processes [46]. In this direction, Wei et al. [47] examined the protective capacity of THC after TBI and found a reduction of oxidative stress caused by brain contusion, reduction of cerebral edema, and reduced death of brain neurons. Different authors observed a curcuminoide-induced reduction in superoxide dismutase (SOD) and glutathione peroxidase (GPx) activities as well as a decrease in MDA levels as markers of oxidative stress after TBI, cardiac damage, and bronchopulmonary dysplasia [48,49,50,51,52,53,54]. In addition, Wei et al. [47] found an increase in the expression of pro-apoptotic factors in comparison to the expression of anti-apoptotic factors, resulting in Bax-dependent pore formation and increased cell permeability, followed by activation of caspase-3 and degradation of DNA and some critical proteins, ultimately resulting in cell death [19]. Administration of 25 mg/kg THC causes suppression of the Bax translocation, upregulating the expression of the anti-apoptotic B-cell lymphoma 2 (Bcl-2) protein. Hence, it seems that the neuroprotective mechanism of THC is based on the blockage of apoptotic mechanisms.

Conversely, Gao et al. [41] found that THC reduces brain edema and improves neurobehavioral function while inhibiting TBI-induced apoptosis, which is mainly activated through autophagy and the phosphatidylinositol 3’-kinase (PI3K)/AKT pathway. Cytoplasmic matter and dysfunctional organelles are sequestered and destroyed in an orderly fashion during the process of autophagy, providing a recycling mechanism for cellular components [55]. Stressful conditions such as starvation [56], subarachnoid bleeding [57], TBI [58], and cerebral ischemia [59] promote autophagy, which subsequently supplies nutrients necessary for the critical maintenance of certain metabolic processes [60]. A study by Gao et al. [41] shows that after TBI, rats treated with THC dramatically increased the activation of the autophagy system, as seen by increased expression of light chain 3 (LC3)-II and beclin-1, as well as decreased expression of p62. However, autophagy after TBI is a double-edged sword, and the mechanisms that regulate its control are unknown [61]. Madathil et al. [62] have shown that the PI3K/AKT signaling pathway plays a key role in the control of cell survival after TBI. At the same time, various neuroprotective drugs such as estradiol [63] and statins [64], through stimulation of the PI3K/AKT pathway, may have therapeutic advantages after TBI. In this direction, Gao et al. [41] found that THC therapy improves AKT phosphorylation, while LY294002, a highly specific PI3K inhibitor, eliminates THC-induced neuroprotection 24 h after TBI. Hence, the PI3K/AKT signaling pathway is probably the most involved mechanism in the anti-apoptotic effect of THC (Figure 5).

Brain damage resulting from a variety of diseases, such as neurodegenerative disorders [65,66,67], and cerebral ischemia [68,69,70] is linked to autophagy. Autophagy’s role in cell survival or death is currently unclear [61]. Clark et al. [71] found that after TBI, there is an intensification of autophagy in human brain tissue and that the oxidative stress that occurs in TBI further exacerbates the neurological damage in mice by altering autophagy [71]. Gao et al. [19] showed that THC therapy increased autophagy and protected the brain from mitochondrial apoptosis in a rat model of TBI.

CUR has been proven to have a neuroprotective effect against brain injury caused by cerebral ischemia/reperfusion (I/R) [72]. Tyagi et al. [73] found that THC lowers infarction by improving neurological outcomes after I/R injury in CBS heterozygous knockout mice with hereditary hypercysteinemia (HHcy). THC reduces cytochrome c homocysteinylation by decreasing oxidative stress and matrix metalloproteinase 9 (MMP9), as well as protecting neurons via autophagic mechanisms. [9]. The BBB could be disrupted in a variety of clinical situations, including I/R injury, which causes increased vascular permeability and the formation of cerebral edema [74]. Tyagi et al. [73] discovered that THC lowers homocysteine (Hcy) neurotoxicity by decreasing endothelial cell damage, which is the reason for BBB integrity preservation. Hcy-induced N-homocysteinylation induces protein structural disruption, which leads to vascular injury [74,75,76]. Tyagi et al. [73] studied the impact of Hcy on the homocysteinylation of cytochrome c following I/R damage and discovered that THC reduced the homocysteinylation of cytochrome c by lowering oxidative stress, which leads to MMP-9 activation [76]. MMP-2 and MMP-9 develop early in HHcy and are related to cardiovascular and neurovascular diseases [77,78,79]. Thus, MMP-9 is involved in the pathological proteolytic breakdown of the BBB, and its enhanced activation is linked to brain dysfunction caused by I/R damage [79,80]. Tyagi et al. [81,82], on the other hand, found that Hcy induces apoptosis or autophagy/mitophagy. Adhami’s group [68,83] found that upon I/R injury in mice, many damaged neurons show autophagic/lysosomal cell death features. Otherwise, the study by Ventruti et al. [84] implies a relationship between autophagy and neuroprotection. Hence, the association between autophagy and cell death or survival during cerebral I/R remains an enigma. Tyagi et al. [82] found that in genetic HHcy mice, THC treatment improves autophagy after cerebral I/R injury. It appears that autophagy may be a unique method by which persistent ischemic stroke induces neuronal death, and its inhibition may help reduce the damage from I/R injury.

In addition, Zhan et al. [85] discovered that in diseased circumstances, growth receptor-induced ERK signaling influences proliferation and differentiation. Redox imbalance, brain ischemia, and neurotransmitter release may all activate these pathways [86]. Lin et al. investigated the expression of Golgi reassembly-stacking protein of 65 kDa (GRASP65) and phosphorylated-GRASP65 (pGRASP65), which are membrane proteins involved in Golgi-stacking, cell division, proliferation, and apoptosis [87,88,89,90,91,92]. Extensive experimental data have demonstrated that GRASP65 is a substrate of ERK as well as of cyclin-dependent kinase 1 (CDK1) and polo-like kinase 1 (PLK1), and the action of these kinases is responsible for the depolymerization and division of the Golgi apparatus during mitosis [88,89,91,92,93]. THC administration, according to Lin et al. [87], resulted in a dose-dependent reduction in ERK-mediated GRASP65 phosphorylation (Figure 6). Under conditions of high oxidative stress, the Golgi apparatus, as a downstream target organelle associated with GRASP65 phosphorylation, is essential for the endoplasmic reticulum and mitochondria. The Golgi apparatus’s reaction to stress restricts the production of critical proteins, which undoubtedly influences the severity of I/R damage [87]. The same authors observed that THC administration attenuated I/R damage-induced SOD depletion dose-dependently. THC therapy also lowers MDA elevations caused by I/R injuries in a dose-dependent manner [87].

## 6. Anti-Amyloid Activity of THC

The use of CUR to treat Alzheimer’s disease (AD) has sparked considerable interest due to its powerful anti-amyloid and anti-inflammatory characteristics, since this polyphenol is less toxic and less costly than most other therapies [94,95,96,97,98]. Most studies emphasize the anti-amyloid activities of CUR in turmeric extract; however, it also includes a high concentration of other polyphenols, including BDMC and DMC [97]. In addition, these compounds are metabolized in the liver and produce significant amounts of a relatively stable, water-soluble metabolite, namely THC. To determine the anti-amyloid properties of THC, Maiti et al. [99] compared the binding and aggregation inhibition efficiency of CUR, BDMC, DMC, and THC (Figure 7) in relation to Alzheimer’s Aβ42 and Aβ40 peptides.

The authors found that DBMC, DMC, and THC had a stronger interaction with Aβ40 and Aβ42. The same scientists reported that the majority of the chemicals favored binding in the N-terminal sequence of Aβ’s core hydrophobic region, indicating that this binding is responsible for inhibition of Aβ aggregation. Maiti et al. [99] observed that keto-CUR (KCUR) has the lowest binding energy for CUR derivatives and Aβ40 and Aβ42, suggesting that KCUR has a greater binding affinity to both Aβ40 and Aβ42 than other CUR derivatives, followed by enol-CUR (ECUR), BDMC, DMC, and THC (Figure 7) [99]. As a result, they demonstrated that in the presence of CUR derivatives, the two Aβ molecules dissociate from clumping together. 

The higher affinity of KCUR for Aβ is due to its lipophilicity, which allows it to penetrate the hydrophobic core of Aβ aggregates, preventing further aggregation. THC, on the other hand, is projected to form weaker contacts with the hydrophobic residues of Aβ owing to its greater hydrophilicity; nonetheless, due to its high stability, it inhibits Aβ aggregation to a comparable extent as KCUR or other CUR derivatives [99].

Considering that CUR can form H-bonds with a variety of Aβ-amino acid residues, primarily N-terminal or occasionally C-terminal amino acids [100], Maiti et al. [99] examined the binding energy between Aβ’s binding pocket and various amino acids and discovered a favorable interaction between a greater number of Aβ’s amino acids and ECUR or THC [101]. The same researchers examined the number of CUR-derivative molecules required to induce certain effects during Aβ aggregation and concluded that a minimum of 12–18 CUR molecules are required to significantly reduce aggregation, while in the case of THC, the minimal number of molecules is between 5–6, indicating that THC has a more significant Aβ42 inhibitory effect than CUR.

To further investigate the neuroprotective effects of CUR and THC, Maiti et al. [99] measured protein kinase B (Akt) and caspase-3 levels in Aβ42-treated SH-SY5Y neuroblastoma cell cultures and observed that both CUR and THC (1 mmol/L) significantly reduced caspase-3 levels and caused an increase in the level of Akt, suggesting that both compounds may prevent apoptotic death (Figure 5). Further investigation of the effects of THC on the induction of molecular chaperones such as heat shock proteins (HSPs) showed that different concentrations of THC induced HSP90 and HSP70 levels in SH-SY5Y cells, similar to CUR treatment, suggesting that THC plays a significant role in protein quality control and inhibition of Aβ aggregation [102], which has also been observed in the case of other CUR derivatives. The molecular mechanisms of HSP induction by CUR derivatives and/or by THC are not clear yet, although Maiti et al. [99] confirmed that THC could induce a CUR-like HSP response.

## 7. THC-Related Inhibition of Cell Cycle Arrest and Apoptosis in Microglia through Ras/ERK Signaling

The study of Xiao et al. [103] showed that THC treatment of BV-2 cells (microglial cells immortalized by v-raf/v-myc carrying J2 retrovirus and expressing nuclear v-myc and the cytoplasmic v-raf oncogene products as well as the env gp70 antigen at the surface level) exposed to Aβ can alleviate the reduced cell viability and inhibit cell cycle arrest and apoptosis. A comprehensive proteomic analysis of hippocampal tissue from APP/PS1 mice (double-transgenic mice expressing a chimeric mouse/human amyloid precursor protein and a mutant human presenilin 1) revealed that the effects of THC in controlling the development of amyloid plaques are related to the activation and regulation of immune cells. Meanwhile, proteomic analyses by Xiao et al. [103] suggest that THC-induced suppression of Ras and JAK–STAT signaling pathways is involved in cell progression. The Ras/ERK signaling pathway is the main controller of cell survival, differentiation, proliferation, metabolism, and motility during extracellular induction [104] and is directly related to the G1/S transition in the cell cycle [105]. K-Ras activation induces upregulation of several cell cycle stimulators, such as cyclin D, which accelerates the G1/S transition [106]. The *Ccnd2* gene encodes a specific G1/S cyclin-D2, which functions as a regulatory subunit of CDK4 and CDK6, whose activity is required for the G1/S cell cycle transition [106,107]. CDKN1A, known as a cyclin-dependent kinase inhibitor 1A, prevents the phosphorylation of critical cyclin-dependent kinase substrates and different signaling pathways involved in the transcriptional activation of the *Cdkn1a* gene [108,109]. Xiao et al. [103] found that Aβ downregulated the expression of Grb-associated binding 2 (GAB2) and K-Ras proteins and inhibited the transcriptional expression of *Ccnd2* and *Cdkn1a* genes in BV-2 cells. Decreased expression of Gab2 and K-Ras in vivo is also observed in APP/PS1 mice. Precisely, Xiao et al. [103] confirmed that THC treatment causes attenuation of the up-regulated expression of GAB2 and K-Ras in APP/PS1 mice in addition to BV-2 cells exposed to Aβ. THC treatment also caused alleviation of the down-regulated *Ccnd2* gene induced by Aβ in BV-2 cells, suggesting that THC generally attenuates Aβ-induced G1/S arrest in BV-2 cells via the Ras/ERK signaling pathway. However, the data also show that THC did not affect the Aβ-induced upregulation of *Cdkn1a*. It has to be pointed out here that the mechanism underlying Aβ-induced upregulation of *Cdkn1a* transcription and the relationship between decreased *Cdkn1a* transcription and oligomeric Aβ-induced cell cycle arrest and apoptosis require further investigation. 

Considering that cell cycle progression and apoptosis are closely related, Xiao et al. [103] examined the expression of caspase-3, poly [ADP-ribose] polymerase 1 (PARP1), and cleaved-PARP1 to check apoptosis. In doing so, they found that inhibition of PARP-1 could prevent Aβ-induced neuronal death [110]. Furthermore, during the apoptotic process, caspase-3 induces the cleavage of PARP1 into an 85–89 kDa COOH-terminal fragment [111]. The appearance of PARP1 fragments is commonly considered an important biomarker of apoptosis. Xiao et al. [103] showed that Aβ upregulates the expression of caspase-3, PARP1, and cleaved-PARP1 in BV-2 cells, while upregulation of caspase-3 was also observed in APP/PS1 mice. THC treatment down-regulated caspase-3 in APP/PS1 mice and Aβ-exposed BV-2 cells and decreased the expression of PARP1 and cleaved-PARP1 in Aβ-exposed BV-2 cells, indicating a summative effect of THC in inhibiting Aβ-induced apoptosis.

On the other hand, Bcl-2-associated athanogene 1 (Bag1) (cochaperone for the heat-shock protein Hsp70 that interacts with C-Raf, B-Raf, Akt, Bcl-2, steroid hormone receptors, and other proteins) as another potential THC-affected player possesses several functions, among which the most important are the activation of Raf-1 (proto-oncogene serine/threonine-protein kinase) through its N-terminal domain that promotes cell growth [112,113] and binding to Bcl-2 (a cellular protein that inhibits apoptosis), which enhances the antiapoptotic activity of Bcl-2 [114]. Actually, Xiao et al. [103] found that THC up-regulates Bag1 expression in APP/PS1 mice and BV-2 cells exposed to Aβ, suggesting the combinatorial effects of THC on inhibition of cell cycle arrest and apoptosis by up-regulating Bag1.

Higher expression of TNF-α and TGF-β1 as concomitant AD mechanisms is reported in APP/PS1 mice, while THC administration reduces TNF-ɑ and up-regulates TGF-β1 expression in APP/PS1 mice [103]. In BV-2 cells, Aβ induces up-regulation of TNF-ɑ and down-regulation of TGF-β1, indicating that Aβ accumulation generally induces a more self-sustaining inflammatory reaction than an increased phagocytic capacity of BV-2 cells under experimental conditions. The down-regulation of TNF-α and up-regulation of TGF-β1 by THC suggest that the effects of THC on neuroprotection probably involve alternative activation of microglia, which warrants further investigation (Figure 8).

## 8. CUR- and THC-Associated Effects on Parkinson’s Disease Progression

Rajeswari et al. [20] investigated the effects of CUR and THC on the progression of Parkinson’s disease (PD). According to their results, CUR and THC normalized the depletion of dopamine (DA) and 3,4-dihydroxyphenylacetic acid (DOPAC) caused by 1-methyl-4-phenyl-1,2,3,6-tetrahydropyridine (MPTP) and also had a considerable impact on the activity of monoamine oxidase (MAO) B (MAO-B) in the striatum. MPTP’s activity mainly affects the nigrostriatal system [115]. Upon its administration, MPTP quickly crosses the BBB and is converted to the 1-methyl-4-phenyl pyridinium ion (MPP^+^) via the action of MAO in the brain. In turn, dopamine transporters are responsible for the selective transport of MPP^+^ into dopaminergic neurons [115]. Its subsequent accumulation in the mitochondria [116,117,118] leads to increased ROS production, which is toxic to neurons [119,120]. The inhibition of MAO-B by CUR and THC resulted in an increase in DA and DOPAC levels [20]. CUR was also found to increase DA levels in the frontal cortex and striatum and inhibit brain MAO-B activity in the 6-OHDA animal model of PD [121]. All these findings emphasize the neuroprotective effects of CUR and THC treatment in the direction of MAO-B inhibition and preservation of DA and DOPAC levels. Thus, according to Rajeswari et al. [20], CUR’s and THC’s inhibitory effects on MAO-B could offer significant benefits in slowing the progression of PD. 

## 9. Conclusions

Numerous recently published in vitro and in vivo studies show that the application of THC can prevent the occurrence of various diseases related to oxidative disorders, primarily due to its strong antioxidant activity. Research shows that THC reduces the biochemical, neurophysiological, and histological changes caused by vincristine treatment in rats. The obtained results indicated that the benefits of THC in such processes might be related to different mechanisms, including antinociceptive, anti-inflammatory, Ca^2+^-accumulation-inhibitive, TNF-α-suppressive, neuroprotective, and antioxidant activities. THC therapy also has a neuroprotective effect on TBI-induced apoptosis, potentially through autophagy and induction of the PI3K/AKT pathway. As a result of these findings, THC may be a beneficial therapeutic agent for TBI therapy.

Furthermore, if HHcy is proven to cause neurodegenerative disorders (stroke), THC may be an effective prophylactic agent in preventing Hcy-induced oxidative stress. THC also shows a protective effect against damage caused by cerebral I/R, which is probably mediated by inhibition of the ERK signaling pathway and subsequent reduction of GRASP65 phosphorylation. Based on this, THC may be a useful therapeutic agent to prevent brain I/R-induced damage.

Both in silico and in vitro data suggest that THC has anti-amyloid and neuroprotective properties similar to those of CUR. THC, being a more stable metabolite of CUR, has the potential to more effectively inhibit Aβ aggregation than other CUR derivatives. Nonetheless, further investigation is needed, particularly using various animal models of AD, to verify these results and optimize them for future therapeutic use. The identification of THC’s influence on amyloid plaque development in a mouse model of AD, which has a vital role in restoring cell cycle homeostasis, as well as THC’s inhibitory effects on microglia apoptosis via the Ras/ERK signaling pathway, provide fresh insights on the potential of THC in slowing the progression of AD.

However, as with other natural compounds, based on its limited pharmacokinetic properties, the applicability of THC as a lead compound could depend on an appropriate formulation bypassing the first-pass metabolism. Further improvement of the bioavailability of THC in vivo is a key direction for future research. Combinatorial therapies that target multiple processes. such as reducing oxidative stress and enhancing anti-inflammatory effects. may offer greater opportunities for clinically meaningful prevention. In addition, with the study of Pari and Murugan (2006) [6] in mind, besides its application in combination with other compounds for better effects, special attention should be given to its doses.

## Figures and Tables

**Figure 1 molecules-28-03734-f001:**
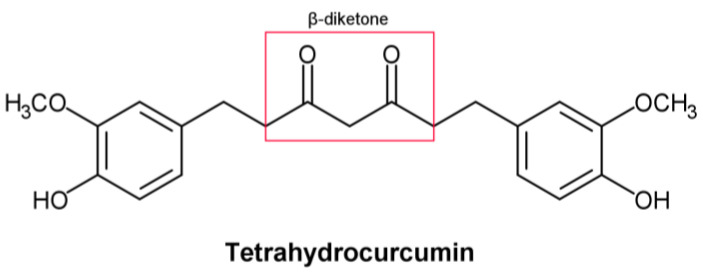
Structural formula of tetrahydrocurcumin.

**Figure 2 molecules-28-03734-f002:**
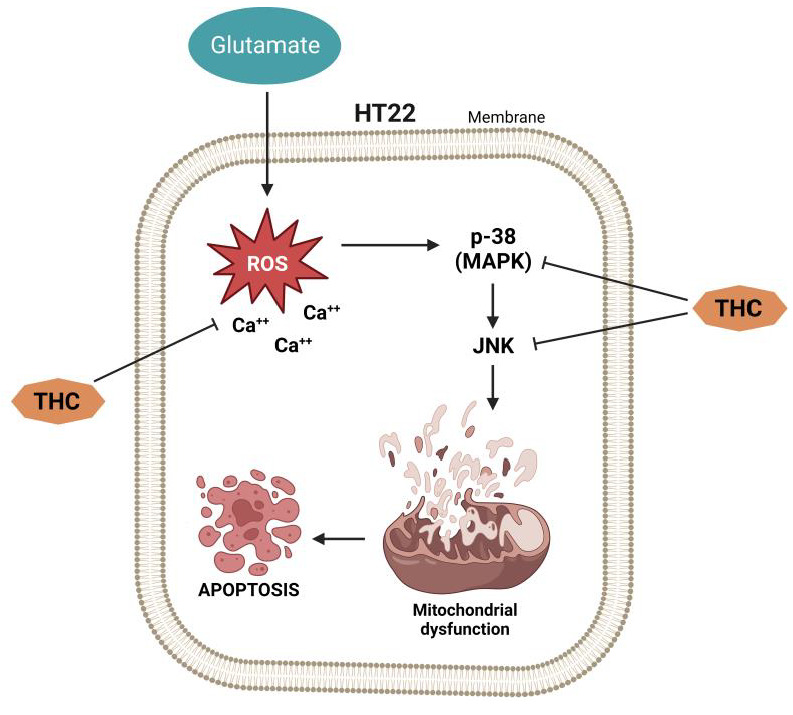
The effect of THC on glutamate-induced apoptotic damage in HT22 cells.

**Figure 3 molecules-28-03734-f003:**
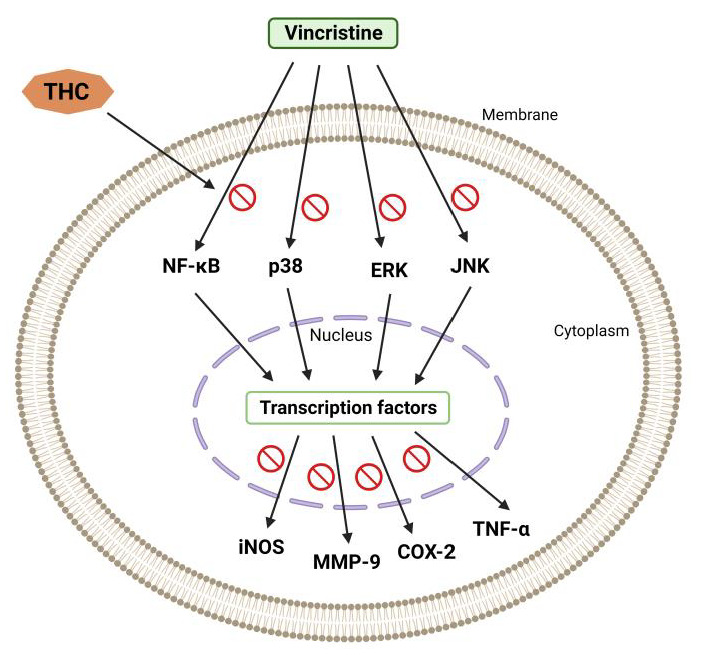
Vincristine-induced peripheral neuropathy. THC-induced suppression of the nuclear factor kappa light chain enhancer of activated B cells (NF-κB), c-Jun N-terminal kinase (JNK), p38, extracellular signal-regulated kinase (ERK), inducible NO synthase (iNOS), cyclooxygenase 2 (COX-2), and matrix metalloproteinase (MMP-9).

**Figure 4 molecules-28-03734-f004:**
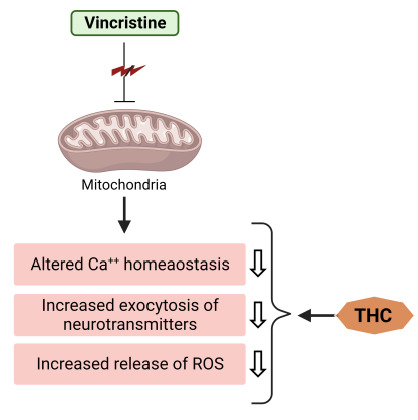
THC attenuates vincristine-induced pathogenesis at the level of mitochondria.

**Figure 5 molecules-28-03734-f005:**
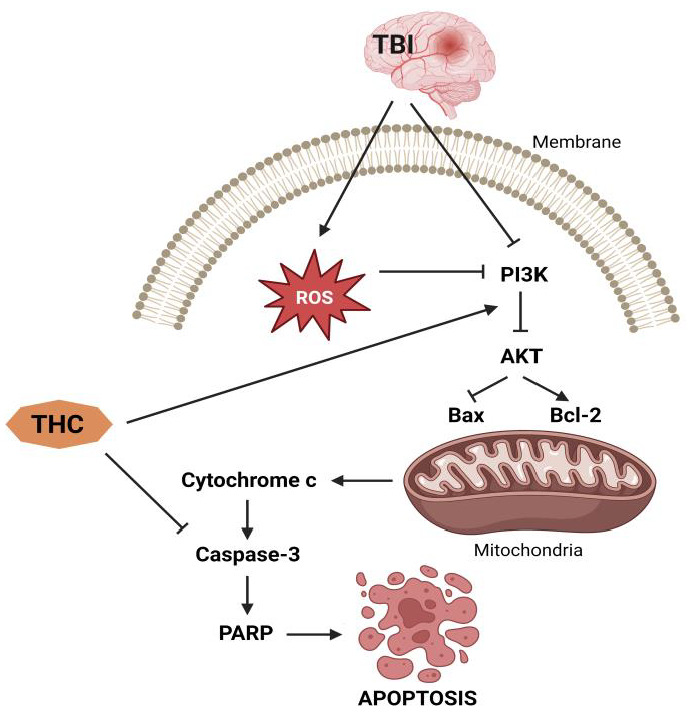
Anti-apoptotic effects of THC.

**Figure 6 molecules-28-03734-f006:**
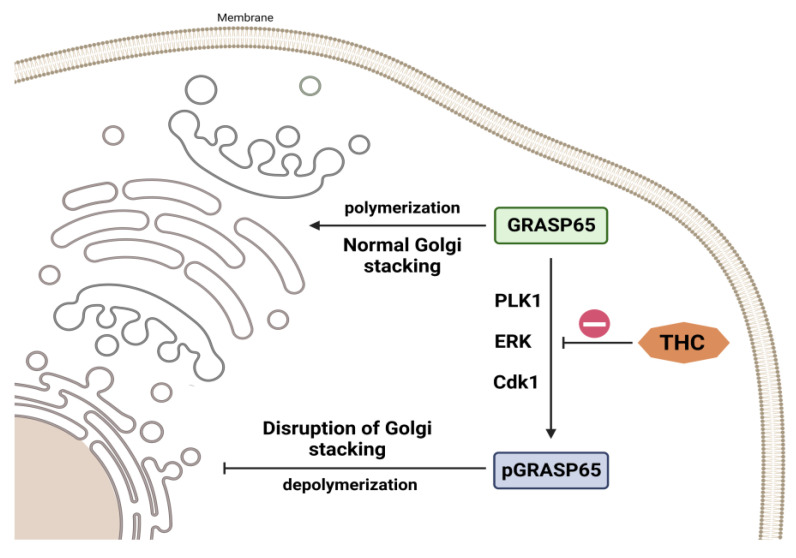
THC impact on Golgi stacking. THC causes a dose-dependent decrease in ERK-mediated GRASP65 phosphorylation.

**Figure 7 molecules-28-03734-f007:**
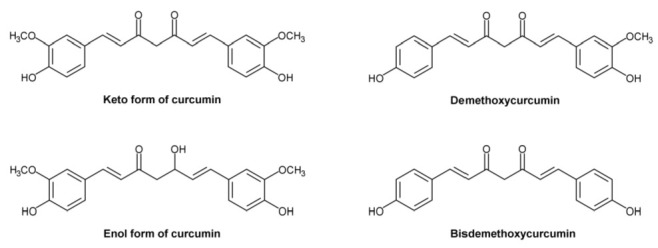
Structural formulas of ketocurcumin, enolcurcumin, demethoxycurcumin, and bisdemethoxycurcumin.

**Figure 8 molecules-28-03734-f008:**
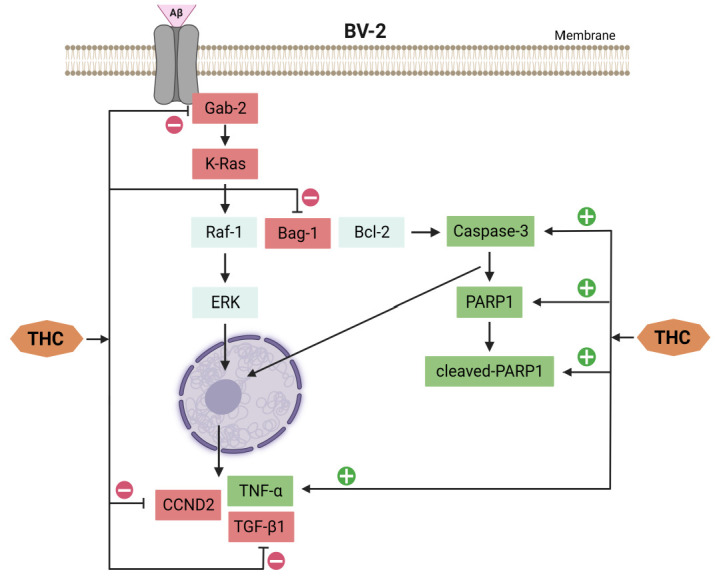
THC affects signaling pathways for inhibition of cell cycle arrest and apoptosis induced by Aβ in BV-2 cells.

## Data Availability

Not applicable.

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
