# Peer review of "Positive Tetrahydrocurcumin-Associated Brain-Related Metabolomic Implications"

_molecules, 2023, doi:10.3390/molecules28093734_

Round 1
Reviewer 1 Report
The paper described a review of the preventive role of tetrahydrocurcumin (THC) on various brain dysfunctions and processes, as well as, suggested THC as a dietary supplement to protect against traumatic brain injury and improve brain function.
The review is well-organized and three modifications are suggested to authors:
1. The authority of the name plant can not be written in italics, as shown in line 38.
2. In lines 44 to 49 the chemical features of the structures were described. A new figure representing the compounds should be added in the paper.
3. There is no deeper critical assessment of data, and as submitted it is an encyclopaediac presentation of the topic . From my point of view, a good review should not only list facts but should also provide deeper interpretation and guide further work in the field. Therefore, the authors should to explore the topic to greater depth.
Author Response
Dear Editor,
First of all, we would like to thank you very much for all efforts that you made to improve the quality of our paper. Also, we appreciate all the suggestions and requirements of the reviewers and want to express our sincere gratitude for their valuable evaluation of the manuscript. We hope that this new corrected form of the manuscript will completely cover all requirements and you will find it suitable for publication.
Also, I would like to mention that we included two new coauthors (our colleagues from China), which comments we found were very helpful and impactful on the quality of the manuscript.
Responses to the reviewer’s questions:
Review 1:
Comments and Suggestions for Authors
The paper described a review of the preventive role of tetrahydrocurcumin (THC) on various brain dysfunctions and processes, as well as, suggested THC as a dietary supplement to protect against traumatic brain injury and improve brain function.
The review is well-organized and three modifications are suggested to the authors:
- The authority of the named plant cannot be written in italics, as shown in line 38.
Answer: The suggestion is accepted and appropriate corrections are included in the manuscript
- In lines 44 to 49, the chemical features of the structures were described. A new figure representing the compounds should be added to the paper.
Answer: The suggestion is accepted and two new figures (Fig. 1 and Fig.7), representing the structure of the compounds are included in the manuscript.
- There is no deeper critical assessment of data, and as submitted it is an encyclopaediac presentation of the topic. From my point of view, a good review should not only list facts but should also provide deeper interpretation and guide further work in the field. Therefore, the authors should explore the topic in greater depth.
Answer: We provided new paragraphs that rely on the new metabolomics and pharmacokinetic characteristics of the compound. Based on that new avenues for investigation were directed and we expect that in this form, the article will attract broader attention.
(Please have a look at Pg.2, from line 70 to line 83, and Pg.3, from line 89 to line 100).
Sincerely yours,
Mitko Mladenov,

Reviewer 2 Report
Tetrahydrocurcumin is a natural product from curcumin. Josifovska et al. reviewed the neuro-related function of tetrahydrocurcumin and tried to explain the mechanisms of action. This work is very intriguing, nevertheless, I have some suggestions to make regarding the manuscript:
Major issues:
1, Please provide the structure of THC in the manuscript.
2, The authors mainly focused on the studies of cell lines or animals. How about the pharmacokinetics studies?
3, Line 404, the authors mentioned that THC had a “limited oral bioavailability”. What is the progress in pharmaceutical preparations? Maybe the dosage form may affect the efficacy of THC via different mechanisms.
4, A review is a paper written by summarizing, analyzing, and refining a large number of original research papers on a certain topic at a certain time. Of 115 references, there are more than 20 references were published 20 years ago, and only 6 references were published after 2020. Either the research on THC seemed slowed down in recent years or the authors failed to present the latest research progress, the scientific value of the review is low. Furthermore, this review failed to analyze the previous studies and provided limited guiding significance for further study.
Minor issues:
1, Line 151, [[38]].
Author Response
Dear Editor,
First of all, we would like to thank you very much for all efforts that you made to improve the quality of our paper. Also, we appreciate all the suggestions and requirements of the reviewers and want to express our sincere gratitude for their valuable evaluation of the manuscript. We hope that this new corrected form of the manuscript will completely cover all requirements and you will find it suitable for publication.
Also, I would like to mention that we included two new coauthors (our colleagues from China), which comments we found were very helpful and impactful on the quality of the manuscript.
Responses to the reviewer’s questions:
Review 2:
Comments and Suggestions for Authors
Tetrahydrocurcumin is a natural product from curcumin. Josifovska et al. reviewed the neuro-related function of tetrahydrocurcumin and tried to explain the mechanisms of action. This work is very intriguing, nevertheless, I have some suggestions to make regarding the manuscript:
Major issues:
- Please provide the structure of THC in the manuscript.
Answer: The suggestion is accepted and a completely new Fig. 1 is included in the manuscript.
- The authors mainly focused on the studies of cell lines or animals. How about the pharmacokinetics studies?
Answer: the suggestion is accepted and a completely new paragraph concerning the pharmacokinetic properties of THC concerning other curcuminoids is discussed.
(Please have a look at Pg.2, from line 87 to Pg.3, line 96).
- Line 404, the authors mentioned that THC had a “limited oral bioavailability”. What is the progress in pharmaceutical preparations? Maybe the dosage form may affect the efficacy of THC via different mechanisms.
Answer: The main focus of the manuscript was to point out brain-related characteristics of THC. But still, the importance of plasma THC concentration was mentioned in a few places in the manuscript. In addition, in references 6, 7, and 8 from our Reference list, the effects of different doses of THC were discussed. We cited these references in the body text but didn’t pay special attention to them, because the basic focus of the manuscript was not related to the treatment. Anyway the dosage of 80 mg/kg body weight) was found to significantly reduce oxidative damage [6], which was the reason why we mentioned this dose. At the end of the manuscript in the Conclusion section, you can find an additional explanation that besides its application in combination with other compounds for better effects special attention should be given to its doses.
(Please have a look at Pg.2, from line 64 to line 68, and Pg.12, from line 443 to line 452).
- A review is a paper written by summarizing, analyzing, and refining a large number of original research papers on a certain topic at a certain time. Of 115 references, there are more than 20 references were published 20 years ago, and only 6 references were published after 2020. Whether the research on THC seemed slowed down in recent years or the authors failed to present the latest research progress, the scientific value of the review is low. Furthermore, this review failed to analyze the previous studies and provided limited guiding significance for further study.
Answer: To fulfill reviewer remarks we included 6 new references concerning the pharmacokinetic characteristics of THC, and discuss its properties based on which we draw a new manner for its clinical use. So, now the total number of references is 121, and a few of them are published in 2022.
Minor issues:
1, Line 151, [[38]].
Answer: the suggestion is accepted:
Sincerely yours,
Mitko Mladenov,

Round 2
Reviewer 1 Report
Dear Authors,
The suggested revisions were done in the paper and the final version was improved at all. I recommend the publication of the paper.
Reviewer 2 Report
Well done! Looking forward to the publication of your manuscript.